# Efficient Separation of Photoexcited Charge at Interface between Pure CeO_2_ and Y^3+^-Doped CeO_2_ with Heterogonous Doping Structure for Photocatalytic Overall Water Splitting

**DOI:** 10.3390/ma14020350

**Published:** 2021-01-12

**Authors:** Honghao Hou, Hirohisa Yamada, Atsumi Nitta, Yoshinori Murakami, Nobuo Saito

**Affiliations:** 1Department of Materials Science and Technology, Nagaoka University of Technology, Nagaoka 940-2188, Japan; wily9999@outlook.com; 2Department of Chemical Engineering, National Institute of Technology Nara College, Nara 639-1080, Japan; yamada@nara.kosen-ac.jp; 3Department of Environmental Materials, National Institute of Technology Niihama College, Niihama 792-8580, Japan; anitta@mat.niihama-nct.ac.jp; 4Department of Materials Engineering, National Institute of Technology Nagaoka College, Nagaoka 940-8532, Japan; murakami_mb@nagaoka-ct.ac.jp

**Keywords:** overall water splitting, Y^3+^-doped CeO_2_ photocatalyst, charge separation at interface, heterogeneous doping structure

## Abstract

Enhancement of photoexcited charge separation in semiconductor photocatalysts is one of the important subjects to improve the efficiency of energy conversion for photocatalytic overall water splitting into H_2_ and O_2_. In this study, we report an efficient separation of photoexcited charge at the interface between non-doped pure CeO_2_ and Y^3+^-doped CeO_2_ phases on particle surfaces with heterogeneous doping structure. Neither non-doped pure CeO_2_ and homogeneously Y^3+^-doped CeO_2_ gave activities for photocatalytic H_2_ and O_2_ production under ultraviolet light irradiation, meaning that both single phases showed little activity. On the other hand, Y^3+^-heterogeneously doped CeO_2_ of which the surface was composed of non-doped pure CeO_2_, and Y^3+^-doped CeO_2_ phases exhibited remarkable photocatalytic activities, indicating that the interfacial heterostructure between non-doped pure CeO_2_ and Y^3+^-doped CeO_2_ phases plays an important role for the activation process. The role of the interface between two different phases for activated expression was investigated by selective photo-reduction and oxidation deposition techniques of metal ion, resulting that the interface between two phases become an efficient separation site of photoexcited charge. Electronic band structures of both phases were investigated by the spectroscopic method, and then a mechanism of charge separation is discussed.

## 1. Introduction

Hydrogen is a clean and sustainable carbon neutral fuel with oxygen. In order to make hydrogen as manufactured energy, photocatalytic water splitting into H_2_ and O_2_ is considered to be one of the most promising approaches for converting solar energy into hydrogen molecules as storage fuel. Photocatalytic water splitting on semiconductor materials has been extensively studied [1,2,3,4,5,6,7,8] since the photocatalytic water splitting on TiO_2_ photoelectrodes was discovered in 1972 [9]. In the viewpoint of electronic configuration, discovered photocatalysts are classified into two groups: one is transition metal compounds with d^0^ electronic configuration (such as Ti^4+^, Zr^4+^, Nb^5+^, Ta^5+^, W^6+^), the other is typical metal compounds with d^10^ electronic configuration (such as Ga^3+^, In^3+^, Ge^4+^, Sn^4+^ and Sb^5+^) [1,2,3,4,5]. However, most of the discovered photocatalysts still suffer from relatively low efficiency with a low hydrogen/oxygen production rate, which is not enough for practical application. The process of photocatalytic overall water splitting is separated into three major steps: (i) light absorption to generate photoexcited electrons and holes, (ii) transfer photoexcited electron and hole to reaction site, and (iii) water reduction and oxidation on the surface. Among these steps, recombination of photoexcited electrons and holes in step (ii) becomes one of causes of low photoenergy conversion. One way to increase the photoenergy conversion is to prevent recombination of photogenerated electrons and holes [1,2,3,4,5,6,7,8]. To support this purpose, crystal defect reduction in photocatalyst is a promising approach because it acts as recombination center. Recently, Domen et al. discovered that photoenergy conversion significantly increased when well-crystalized SrTiO_3_ synthesized by the flux method was employed [10], indicating that defect reduction is an effective way to prevent the recombination, and then leads to photocatalytic activity enhancement. As another approach to suppress the recombination, it is considered that the artificial structure formation of charge separation on surfaces is an effective method. For instance, Li et al. reported that the interface between Anatase-TiO_2_ and Rutile-TiO_2_, α-Ga_2_O_3_ and β-Ga_2_O_3_ plays an important role for separation of photoexcited charge, and then photocatalytic activity was higher than these of single phase [11,12,13]. The reason why photocatalytic activity enhancement occurs is associated with different electronic structure of both phases. It is considered that photoexcited electron and hole were effectively separated at the interface of both phases because the potential difference of valence band (V.B.) and conduction band (C.B.) act as a driving force of photoexcited charge separation.

CeO_2_ is an important rare earth oxide that has been widely applied in many fields, such as solid oxide fuel cell, heterogeneous catalyst, corrosion prevention, ultraviolet absorber and biomaterial because of nontoxicity and excellent chemical stability [14]. For the application in photocatalytic water splitting, Arakawa et al. have reported that CeO_2_ has an ability of O_2_ evolution from a suspension contained Ce^4+^ as an electron acceptor [15]. On the other hand, it is reported that CeO_2_ suspension can produce H_2_ using S^2−^/SO_3_^2−^ as hole acceptor [16]. The reagent like Ce^4+^ and S^2−^/SO_3_^2−^ are called sacrificial agents. Therefore, it is desirable that photocatalytic water splitting proceed in pure water. However, no photocatalytic ability of CeO_2_ based materials in pure water splitting was reported except for our group, to our best knowledge.

In our previous study, we discovered that Sr^2+^-doped CeO_2_ with f^0^d^0^ electronic configuration became a new photocatalyst for overall water splitting [17]. Furthermore, we found that Sm^3+^-doped CeO_2_ with heterogeneous doping structure has an ability to split water [18]. As a doping metal ion, common characteristics of Sr^2+^ and Sm^3+^ is that ionic radius of doped metal ion were larger than that of Ce^4+^ in fluorite structure, making a stable fluorite lattice based on crystallographic structure [19]. Moreover, as an interesting feature of metal ion doped CeO_2_, neither non-doped pure CeO_2_ and homogeneously Sm^3+^-doped CeO_2_ showed photocatalytic activities under light irradiation. On the other hand, Sm^3+^-heterogeneously doped CeO_2_ of which the surface was composed of non-doped pure CeO_2_ and Sm^3+^-doped CeO_2_ phases exhibited remarkable photocatalytic activities, indicating that the interface between non-doped pure CeO_2_ and Sm^3+^-doped CeO_2_ phases plays an important role for the activation process of photocatalytic water splitting [18]. This discovery was characterized by the charge separation at the interface between two phases gave activity enhancement better than each single phase, but more importantly, activated expression with each inert phase for photocatalytic water splitting reaction. However, a mechanism for charge separation and active site generation at the interface between non-doped pure CeO_2_ and metal ion doped CeO_2_ phases has not been investigated in detail. Therefore, the objective of this study is to confirm whether the interface at both phases become active sites for charge separation. In this study, as a photocatalyst, Y^3+^-doped CeO_2_ was prepared to acquire obvious elemental images at the interface between non-doped pure CeO_2_ and metal ion doped CeO_2_ phase for energy disperse spectroscopic measurement. In case of Sm^3+^-doped CeO_2_ in previous study, it was difficult to get clear elemental images at the interface since an energy of characteristic X-ray emitted from Sm had a value closest to that of Ce. This leads to the hindrance of the investigation on photocatalytic active site using selective photo-reduction and photo-oxidation with metal ions. Therefore, the mechanism of charge separation was not discussed in detail. On the other hand, in case of Y^3+^-doped CeO_2_, an energy of characteristic X-ray emitted from Y was quite different from that of Ce. Furthermore, the since ionic radius of Y^3+^ is larger than Ce^4+^ in CeO_2_ [20,21,22,23,24,25,26], it is expected that Y^3+^-doped CeO_2_ shows efficient photocatalytic activity based on the existence of the interface between non-doped pure CeO_2_ and Y^3+^-doped CeO_2_ phase. In the synthesis of Y^3+^-doped CeO_2_ composed of non-doped pure CeO_2_ and Y^3+^-doped CeO_2_ phase on the surface, a solid state reaction of CeO_2_ and Y_2_O_3_ was used and heated at different calcination temperatures as in a previous report [18]. As a reference, homogeneously Y^3+^-doped CeO_2_ composed of Y^3+^-doped CeO_2_ single phase on the surface was prepared by the co-precipitation method. The doping structures of synthesized Y^3+^-doped CeO_2_ were investigated by X-ray diffraction and energy dispersive spectroscopic methods, and then the relation between the photocatalytic activity and doping structure is discussed. In order to determine the photocatalytic active site, selective photo-reduction of Ru and oxidation deposition of PbO_2_ were employed. In addition, the electronic charge separation mechanism is discussed by results of UV-Vis reflectance spectroscopic and ultraviolet photoelectron spectroscopic measurements.

## 2. Materials and Methods

For photocatalyst preparation by the solid state reaction (referred to as SSR) method, Y^3+^-doped CeO_2_ was prepared by a mixture of CeO_2_ (Nacalai Tesque, 99%) and Y_2_O_3_ (Nacalai) calcinated at different temperatures from 1273 to 1773 K for 16 h under air at atmospheric pressure. After calcination, unreacted residual Y_2_O_3_ was removed by acid treatment with 0.1 M HCl aqueous solution, and then Y^3+^-doped CeO_2_ powder was obtained through filtration of the suspension. The molar amount of dissolved-Y^3+^ in filtrate by the acid treatment was determined by inductively coupled plasma optical emission spectroscopy (ICP-OES; ICPS-7510, SHIMADZU, Kyoto, Japan), and then concentration of doped-Y^3+^ in CeO_2_ at different calcination temperature was calculated by subtracting the molar amount of dissolved-Y^3+^ from that of initial charge. Prepared Y^3+^-doped CeO_2_ in SSR at different calcination temperature are referred to as *x*K-SSR- (*x* represent calcination temperature). As reference sample, pure CeO_2_ was calcined at 1573 K for 16 h under air at atmospheric pressure and is referred to as 1573K-Pure-CeO_2_. For photocatalyst preparation by co-precipitation (referred to as CPT) method, a mixture of Ce(NO_3_)_3_ 6H_2_O (Nacalai Tesque, 99.9%) and Y(NO_3_)_3_ 6H_2_O (Nacalai Tesque, 99.9%) at a molar ratio of Y/(Y + Ce) × 100 = 10.0 mol % was dissolved in distilled water as starting materials. Ammonia aqueous solution (Nacalai Tesque, 28%) was slowly added into prepared solution as a mineralizer with vigorously stirring, and then obtained precursor precipitates were filtrated and dried at 333 K for 12 h under air at atmospheric pressure. Then prepared precursor was calcined at different temperature from 1273 to 1773 K for 16 h under air at atmospheric pressure. Prepared 10 mol % Y^3+^-doped CeO_2_ in CPT at a different calcination temperature is referred to as *x*K-CPT (*x* represent calcination temperature). Furthermore, for the preparation of different Y^3+^-doping concentrations, mixtures of Ce(NO_3_)_3_ 6H_2_O (Nacalai Tesque, 99.9%) and Y(NO_3_)_3_ 6H_2_O (Nacalai Tesque, 99.9%) at a molar ratio of Y/(Y + Ce) × 100 = 0.1, 0.5, 1.0, 5.0, 20.0 and 30.0 mol % were dissolved in distilled water, and precipitates were obtained in a similar fashion. Prepared powders were calcinated at 1573 K for 16 h under air at atmospheric pressure. Prepared samples are referred to as *x*K-CPT-*y*YDC (*x* and *y* represent calcination temperature and additive molar ratio of Y, respectively).

Powder X-ray diffraction pattern was recorded by X-ray diffractometer (XRD; RINT-2000HF with Cu Kα, Rigaku, Tokyo, Japan) to identify crystal structures of prepared sample. Optical property of prepared samples was obtained by ultraviolet-visible diffuse reflectance spectroscopy (DRS; V570, JASCO, Tokyo, Japan), and energy gap (E_g_) was calculated by Tauc-plot. BET surface area was measured by nitrogen adsorption and desorption analyzer (Gemini 2360, Micromeritics, SHIMADZU, Kyoto, Japan). Morphology and particle size were observed by a field emission scanning electron microscope (FE-SEM; SU8230, Hitachi High Technologies, Tokyo, Japan) and a scanning transmission electron microscope (STEM; HT7700, Hitachi High Technologies, Tokyo, Japan). Elemental mapping was performed by FE-SEM and STEM equipped with energy dispersion spectroscopy (EDS; X-Max50, Oxford and Quantax, Bulker, respectively). X-ray photoelectron spectroscopy (XPS; JPS-9010TR, JEOL, Tokyo, Japan) was performed with Mg Kα (*hv* = 1253.6 eV) at 10 kV and 10 mA. The energy-level of VBM (E_VBM_) was investigated by ultraviolet photoelectron spectroscopy (UPS, ESCA 3057, Ulvac-Phi, Tokyo, Japan) with He I excitation (21.2 eV) in 1573K-Pure-CeO_2_ and 1573K-CPT-30.0YDC. UPS spectra were recorded at 9 volts-bias in ultrahigh vacuum chamber.

Photocatalytic water splitting was performed by a homemade-closed gas circulation apparatus directly connected to gas chromatograph equipped with a TCD detector (SHIMADZU GC-390B, MS-5A column, Ar was used as carrier gas, Kyoto, Japan). Prior to the photocatalytic water splitting reaction, the prepared photocatalysts were impregnated up to incipient wetness with Ru_3_(CO)_12_ (Aldrich Chemical Co., 99%, St. Louis, MO, USA) in THF solution. The amount of RuO_2_ was 1.0 wt.% against the prepared sample as Ru meal base. The THF solution was removed under reduced pressure after stirring of the suspension at 333 K for 4 h. The resulting powder was calcinated at 673 K for 4 h under air at atmospheric pressure to oxidize Ru carbonyl complex to RuO_2_ as the promoter. In total, 0.3 g of RuO_2_-loaded photocatalysts were suspended in 150 ml of distilled water. Before the water splitting reaction, dissolved gases in suspension were removed by reducing pressure, and 4.0 kPa of Ar was introduced into the reaction system as an inert circulation gas. Then, UV light (source: 300 W Xe lamp) was irradiated through a Pyrex top-irradiation type vessel with vigorous stirring. The amounts of H_2_ and O_2_ evolution were recorded for 15-min intervals under dark and light irradiation. Wavelength dependence for photocatalytic water splitting was examined by using the same experimental setup, except for cut-off filters (UV35, λ < 350 nm; UV36, λ < 360 nm; L37, λ < 370 nm; L38, λ < 380 nm; L39, λ < 390 nm; L40, λ < 400 nm).

For investigation of the active reduction sites, Ru nanoparticles were photo-deposited in 10.0 vol % methanol and RuCl_3_ aqueous solution under light irradiation. In this suspension, methanol works as a hole scavenger to facilitate the deposition of metallic Ru particles, as reported in the literature [27]. On the other hand, for the investigation of the active oxidation site, photo-deposition of PbO_2_ was performed in a Pb(NO_3_)_2_ aqueous solution under light irradiation. The prepared Ru-deposited and PbO_2_-deposited samples were analyzed by EDS to clarify active sites of reduction and oxidation.

## 3. Results and Discussion

### 3.1. Doping Structures of Y^3+^-Doped CeO_2_ Prepared by SSR and CPT Methods

Figure 1 shows the XRD patterns of Y^3+^-doped CeO_2_ fabricated by SSR and CPT methods. 1573K-Pure-CeO_2_(a) showed a well-crystallized fluorite structure without any impurities. In the case of 1573K-CPT(b), the peak position shifted to a higher angle as shown in the expanded figure from 55.6 to 56.8°, indicating that Y^3+^ was doped in the crystal lattice of CeO_2_ with homogenous doping structures [20,21,22,23]. On the other hand, Y^3+^-doped CeO_2_ prepared by SSR at different temperatures(c)–(h) showed characteristic diffraction patterns corresponding to heterogeneous doping structures. For Y^3+^-doped CeO_2_ prepared by the SSR method, diffraction patterns appeared not only in same position as 1573K-Pure-CeO_2_ but also a higher angle in 1573K-SSR (f) and 1673K-SSR (g), meaning that the prepared sample consisted of the non-doped CeO_2_ and Y^3+^-doped CeO_2_ regions. The intensity of diffraction peaks at a high angle associated with Y^3+^-doped CeO_2_ increased with the increasing calcination temperature. Furthermore, these peaks shifted slightly towards a lower angle with an increasing calcination temperature. However, for 1773K-SSR (h) at the maximum temperature, an unclear diffraction peak for the doping phase was observed, but peak broadening on higher angle side occurred, which indicates that Y^3+^-doped CeO_2_ exists in low concentrations on the surface. Table 1 summarizes the lattice constant of the Y^3+^-doped phase, the percentage of doped Y^3+^ and the percentage of the doped phase obtained by XRD and ICP measurements. Percentages of doped Y^3+^ (doped phase) were calculated from their lattice constant according to references [20,21]. On the other hand, percentages of doped Y^3+^ (Average) were determined by way of ICP measurement as described in the experimental section. For Y^3+^-doped CeO_2_ prepared by the CPT method, the amount of doped Y^3+^ (Doped phase) was the same as that of doped Y^3+^ (average), which means that the prepared Y^3+^-doped CeO_2_ has a homogeneous doping structure as expected. On the other hand, in the Y^3+^-doped CeO_2_ prepared by the SSR method, the amount of doped Y^3+^ (doped phase) was larger than that of doped Y^3+^ (average), indicating that synthesized Y^3+^-doped CeO_2_ was composed of non-doped, lower-doped and/or higher-doped CeO_2_ with heterogeneous doping structures. Since little diffraction peaks corresponding to the Y^3+^-doped phase in 1273K-SSR (f) and 1373K-SSR (g) were observed, surface elemental analysis was performed by using surface sensitive X-ray photoelectron spectroscopic method. As shown in Appendix A, two peaks attributed to Y3d_5/2_ and Y 3d_3/2_ were detected on 1273K-SSR (b) and 1373K-SSR (c) in which little diffraction peaks were observed in XRD measurements. The XPS measurement clearly indicates that all the prepared Y^3+^-doped CeO_2_ have a Y^3+^-doped phase on the topmost surface. Besides, Ce 3d_5/2,_ Ce 3d_3/2_ and their satellite peaks were also detected in all samples. Table 2 shows the Y^3+^ doping concentration calculated from the peak area of Ce 3d_5/2_, Y 3d_5/2_ and the corresponding Relative Sensitivity Factor in X-ray photoelectron spectroscopy (XPS; JPS-9010TR, JEOL) and BET specific surface area. In 1573K-CPT, the percentage of doped Y^3+^ is different from the values determined by XRD and ICP measurements. The reason why the surface Y^3+^ concentration was 2.4 times larger than that of the bulk is not clear, but it is considered that the topmost surface has an imperceptible dopant distribution in the depth direction. The amount of doped Y^3+^ on the surface increased with the increasing calcination temperature, then decreased at 1773K. This phenomenon is in good agreement with the Sm^3+^ doped CeO_2_ in our previous study [18], and it is explained that the calcination procedure at high temperatures gives not only an increase in the doping amount but also dopant diffusion from the doped region to non-doped CeO_2_, showing that the doping structure changes from heterogeneous to homogeneous on the surface.

Figure 2 and Appendix A show FE-SEM /EDS and STEM/EDS images, respectively. In the case of Y^3+^-doped CeO_2_(b) prepared by the CPT method, homogeneous distribution structures of the Y element in CeO_2_ were clearly observed, which are in good agreement with the results of crystallographic analysis described in the previous section. On the other hand, for Y^3+^-doped CeO_2_(c)–(h) synthesized by the SSR method at different calcination temperatures, it was found that the Y element existed in patches, which means Y was heterogeneously doped in CeO_2_. With an increasing calcination temperature, the amount of doped Y increases and then the Y doped area was expanded into non-doped CeO_2_ at a high calcination temperature. This phenomenon is also in good agreement with the results of crystallographic analysis. When focusing on the interface between non-doped CeO_2_ and Y^3+^-doped CeO_2_, the interface area increased with the increasing calcination temperature until nearly 1573 K, reached a maximum of 1573 K, and then decreased due to elemental diffusion from the Y^3+^ doped CeO_2_ to non-doped CeO_2_. In our previous study, in Sm^3+^ doped CeO_2_, we reported that the interface between non-doped CeO_2_ and Sm^3+^-doped CeO_2_ plays an important role for the activation of the photocatalytic reaction [18]. However, as mentioned in the introduction, it was difficult to get clear interface images of SEM/EDS on Sm^3+^-doped CeO_2_. In this study, in the case of Y^3+^-doped CeO_2_, obvious interface images were observed as shown Figure 2e,f.

### 3.2. Investigation of Charge Separation Sites by Photo-Deposition Methods

Figure 3 shows FE-SEM/EDS images after photo-deposition of Ru(a) and PbO_2_(b) on photocatalytic active reduction and oxidation sites, respectively. As shown in Figure 3a, little Ru metal was observed in elemental images of pure-CeO_2_ and Y^3+^ doped CeO_2_ prepared by the CPT method. On the other hand, intriguingly, Ru photo-deposition occurred on Y^3+^ doped CeO_2_ prepared by the SSR method, of which the surface was composed of a heterogeneous doping structure. Ru metal was deposited when the interface of two phases exists on a surface, and they were selectively deposited on the phases of Y^3+^ doped CeO_2_ near the interface. On the other hand, in the case of PbO_2_ deposition as shown in Figure 3b, PbO_2_ was selectively deposited on phases of non-doped CeO_2_ near the interface.

### 3.3. Photocatalytic Activity for Water Splitting on Y^3+^-Doped CeO_2_ with Different Doping Structure

Figure 4 shows the representative photocatalytic water splitting reaction on RuO_2_-loaded Y^3+^-doped CeO_2_ prepared by the SSR method at 1573 K. H_2_ and O_2_ were evolved with a stoichiometric ratio of 2:1 with the light on, which means Y^3+^-doped CeO_2_ became an efficient and stable photocatalyst the same as the Sr^2+^-doped CeO_2_ [17] and Sm^3+^-doped CeO_2_ [18]. Table 3 summarizes the photocatalytic activities for the overall water splitting on the Y^3+^-doped CeO_2_ synthesized by CPT and SSR methods at different calcination temperatures with various doping amounts. Photocatalytic activities were normalized by the specific surface area because specific surfaces decreased with increasing calcination temperatures as shown Table 2. In the RuO_2_-loaded 1573K-Pure-CeO_2_ without Y^3+^ doping, no photocatalytic activity was observed. In addition, RuO_2_-loaded non-doped CeO_2_ with different calcination temperatures at 1273–1773 K showed little activity. In the case of RuO_2_-loaded Y^3+^-doped CeO_2_ synthesized by the CPT method with homogeneous doping structure, photocatalytic activity was not observed at different calcination temperatures with various doping amounts. On the other hand, RuO_2_-loaded Y^3+^-doped CeO_2_ synthesized by the SSR method with a heterogeneous doping structure showed remarkable photocatalytic activity: photocatalytic activity increased with increasing calcination temperatures, reaching a maximum of 1573 K, and then decreased drastically. This phenomenon is in good agreement with our previous reports on Sm^3+^-doped CeO_2_ [18], indicating that the heterogeneous doping structure synthesized by the SSR method plays an important role in creating sufficient photocatalytic activity. As shown in Figure 2 and Appendix A, heterogeneous doping structures composed of non-doped and Y^3+^ doped phases were observed. It seems that the interface region between non-doped and Y^3+^ doped CeO_2_ increased with increasing calcination temperatures, reaching a maximum of 1473–1573 K, and then disappeared at above 1673 K monotonically due to the elemental diffusion from Y^3+^ doped CeO_2_ to non-doped CeO_2_, suggesting that photocatalytic activity was strongly related to the amount of interface between the non-doped and Y^3+^ doped phases. Furthermore, since photo-reduction and oxidation sites appeared near these interfaces, as shown in Figure 3, it is suggested that photocatalytic activity is strongly governed by the interface structure on the surface. It is a well-known fact that CeO_2_ converts photoenergy to thermal energy as used in UV absorbed material, which means that photoexcited electrons and holes are rapidly recombined in the process of nonradiative deactivation. In this study, it was discovered that the interface acts as an effective charge separation site of photoexcited electrons and holes with prevention of the nonradiative deactivation process, and leads to efficient photocatalytic activity for the overall water splitting reaction.

### 3.4. Electronic Band Structures and Schematic Model of Charge Separation at Interface between CeO_2_ and Y^3+^-Doped CeO_2_

Figure 5 shows the UV-Vis diffusion reflectance spectra of 1573K-Pure-CeO_2_(a) and 1573K-CPT-30.0YDC(b), 1573K-SSR(c) and the corresponding Tauc plots(inset). The adsorption edge of the spectrum was approximately 380 nm in 1573K-Pure-CeO_2_(a), on the other hand, shifted towards a higher wavelength in 1573K-CPT-30.0YDC(b) with a homogeneous Y^3+^ doping structure. For 1573K-SSR(c), the spectrum was slightly shifted towards a higher wavelength in comparison with that of 1573K-Pure-CeO_2_(a), indicating that this spectrum consists of two phases of nondoped-CeO_2_ and Y^3+^-doped CeO_2_. As shown in the inserted Tauc plots, the energy gap of 1573K-Pure-CeO_2_(a) and 1573K-CPT-30.0YDC(b) was calculated as 3.30 and 3.20 eV, respectively. Figure 6 shows the excitation wavelength dependence of the photocatalytic H_2_ and O_2_ evolution on RuO_2_-loaded 1573K-SSR by using a cut-off filter. Photocatalytic activities appeared until around 390 nm, and the dependence of photocatalytic activities is in good agreement with the photo-absorption properties measured by UV-vis diffusion refraction spectra, as shown in Figure 5. In the electronic band structure of CeO_2_, the valence band is manly composed of O2p orbital, whereas the conduction band mainly consisted of Ce5d orbital. Furthermore, unoccupied Ce4f orbital is located between O2p and Ce5d orbitals. Since the energy gap from O2p to Ce5d and from O2p to Ce4f orbital is 5.0 and 3.4 eV, respectively [28,29,30,31,32], it is realized that the photocatalytic activity is based on the photo-excitation process from O2p to Ce4f. Figure 7 shows the ultra-violet photoelectron emission spectra of 1573K-Pure-CeO_2_(a), 1573K-CPT-30.0YDC(b) and an enlarged view on top of the valence band. In 1573K-CPT-30.0YDC(b), the spectrum was shifted towards the negative direction compared to that of 1573K-Pure-CeO_2_. As shown in the inserted figure, the valence band maximum (VBM) was determined as −7.5 and −7.2 eV vs. the vacuum level, respectively. For 1573K-CPT-30.0YDC(b), broadening was observed on top of the valence band, suggesting the existence of a slight doping distribution on the surface. Matsui et al. reported that the distance of the Ce-O bond in fluorite-CeO_2_ decreased with increasing Y^3+^ doping concentration [33]. Since the Ce-O distance is strongly related to the energy level of O2p orbital, it is suggested that VBM changes by Y^3+^ doping to CeO_2_. Based on these results, a schematic model of charge separation and efficient overall water splitting at the interface between non-doped and Y^3+^ doped CeO_2_ is shown in Figure 8. Non-doped CeO_2_ and Y^3+^ homogeneously doped CeO_2_ showed negligible photocatalytic activities due to the rapid recombination of the photoexcited electron and hole, but efficient charge separation occurs at the interface between two phases because the potential differences of V.B. and C.B. act as driving forces for suppressing the recombination of the photoexcited electron and hole in the process of nonradiative deactivation.

## 4. Conclusions

RuO_2_-loaded Y^3+^-doped CeO_2_ composed of non-doped and Y^3+^-doped phases with a heterogeneous doping structure exhibited remarkable photocatalytic activity for overall water splitting. Nevertheless, non-doped and homogeneously Y^3+^-doped CeO_2_ showed negligible photocatalytic activity, Y^3+^-doped CeO_2_, of which the surface has an interface of non-doped and Y^3+^-doped phases, showed remarkable photocatalytic activity for overall water splitting. From the selective photo-deposition and spectroscopic measurements, it was found that the interface between non-doped and Y^3+^-doped CeO_2_ plays an important role for suppressing the recombination of photoexcited electrons and holes, and led to a sufficient photocatalytic overall water splitting into H_2_ and O_2_.

## Figures and Tables

**Figure 1 materials-14-00350-f001:**
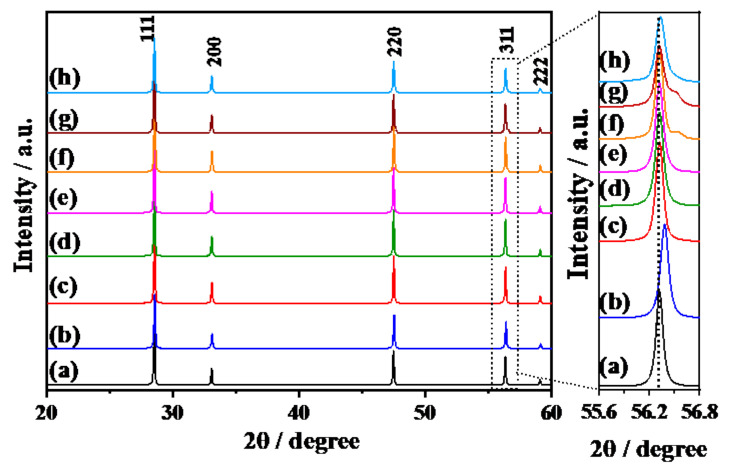
X-ray diffraction patterns of 1573K-Pure-CeO_2_ (**a**), 1573K-CPT (**b**), 1273K-SSR (**c**), 1373K-SSR (**d**), 1473K-SSR (**e**), 1573K-SSR (**f**), 1673K-SSR (**g**), and 1773K-SSR (**h**).

**Figure 2 materials-14-00350-f002:**
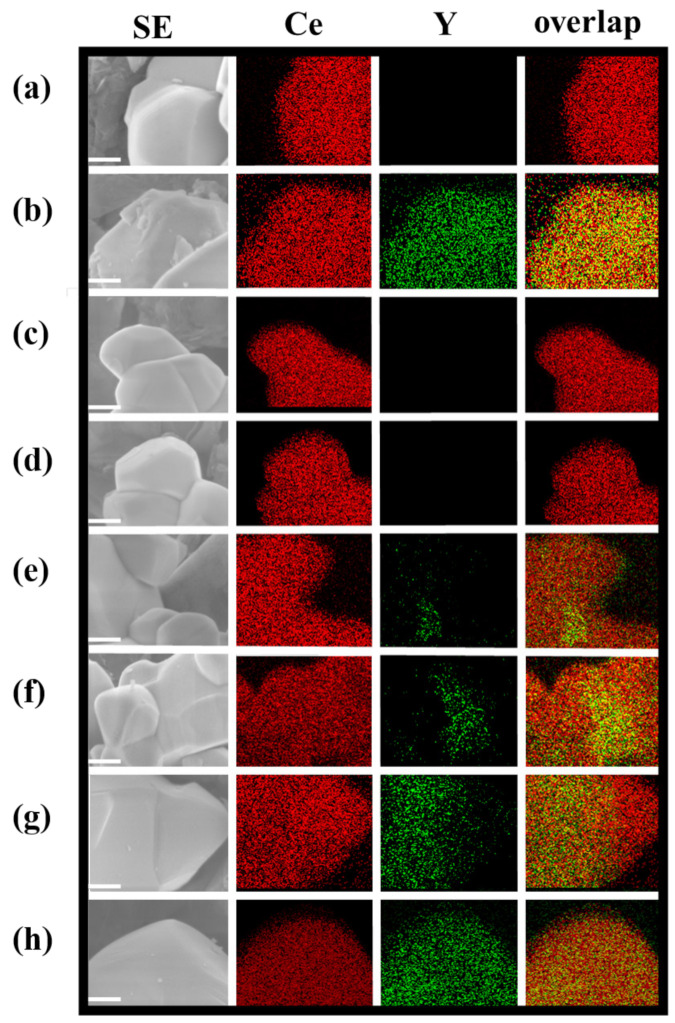
FE-SEM/EDS images of 1573K-Pure-CeO_2_ (**a**), 1573K-CPT (**b**), 1273K-SSR (**c**), 1373K-SSR (**d**), 1473K-SSR (**e**), 1573K-SSR (**f**), 1673K-SSR (**g**), and 1773K-SSR (**h**). Scale bars are 100 nm.

**Figure 3 materials-14-00350-f003:**
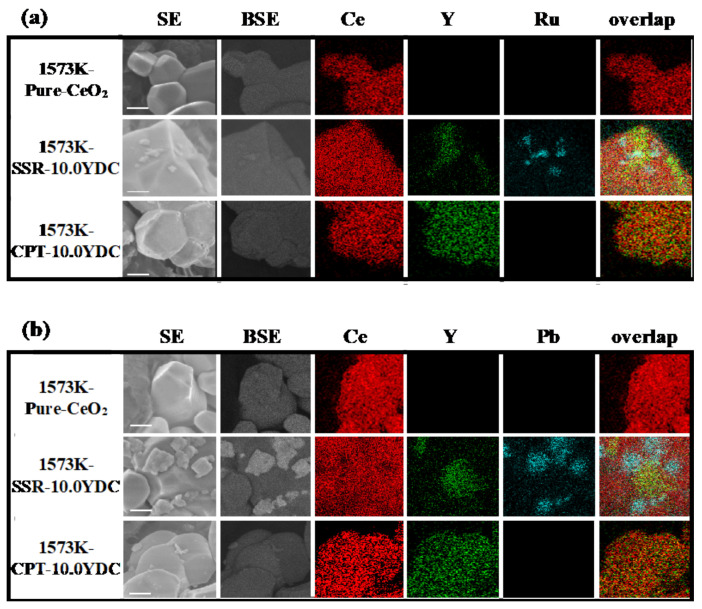
FE-SEM/EDS images of 1573K-Pure-CeO_2_, 1573K-SSR and 1573K-CPT. Ru (**a**) and PbO_2_ (**b**) were photo-deposited on active site. Scale bars are 100 nm.

**Figure 4 materials-14-00350-f004:**
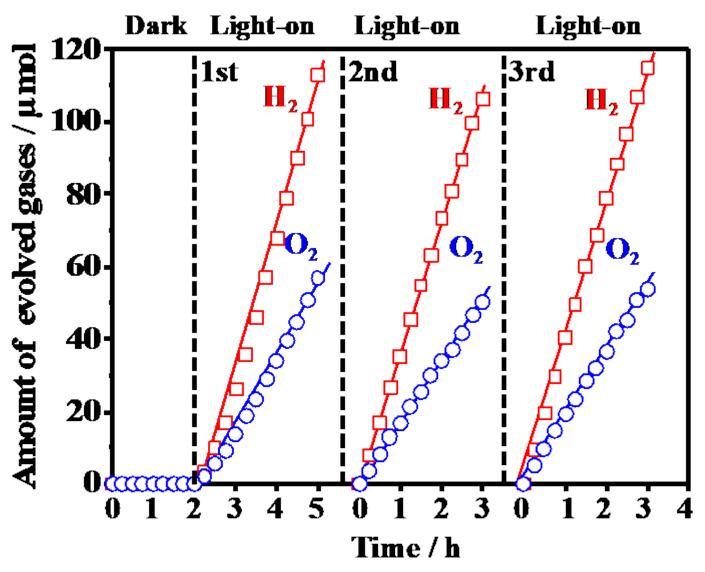
Photocatalytic H_2_ and O_2_ evolution on RuO_2_-loaded 1573K-SSR without and with light irradiation.

**Figure 5 materials-14-00350-f005:**
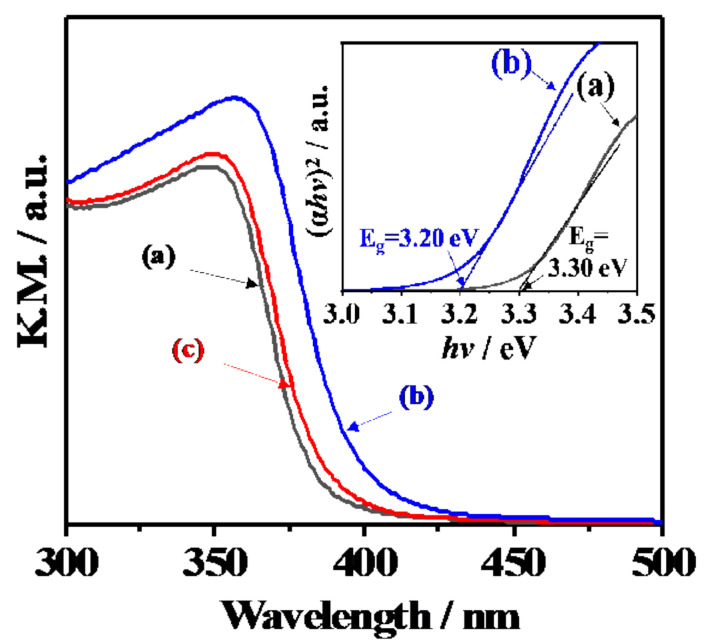
UV-Vis diffuse reflectance spectra of 1573K-Pure-CeO_2_ (**a**), 1573K-CPT-30.0YDC (**b**), 1573K-SSR (**c**) and corresponding Tauc plots(inset).

**Figure 6 materials-14-00350-f006:**
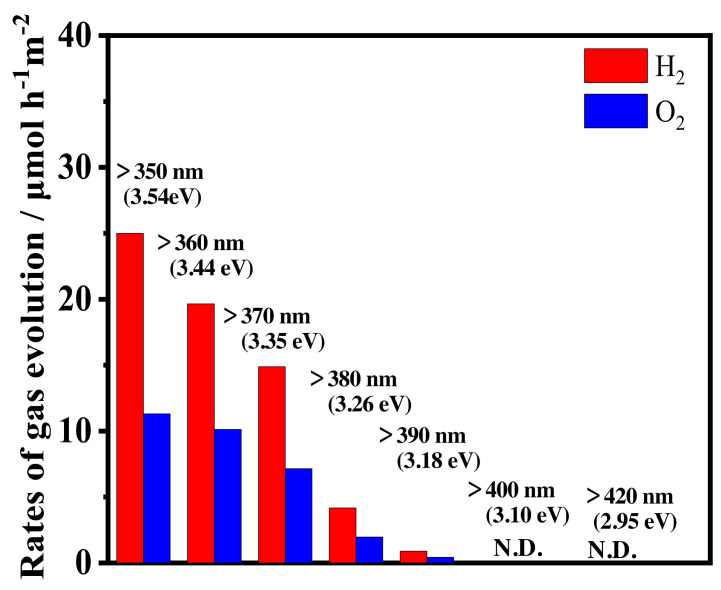
Excitation wavelength dependence of photocatalytic H_2_ and O_2_ evolution on RuO_2_-loaded 1573K-SSR by using cut-off filters.

**Figure 7 materials-14-00350-f007:**
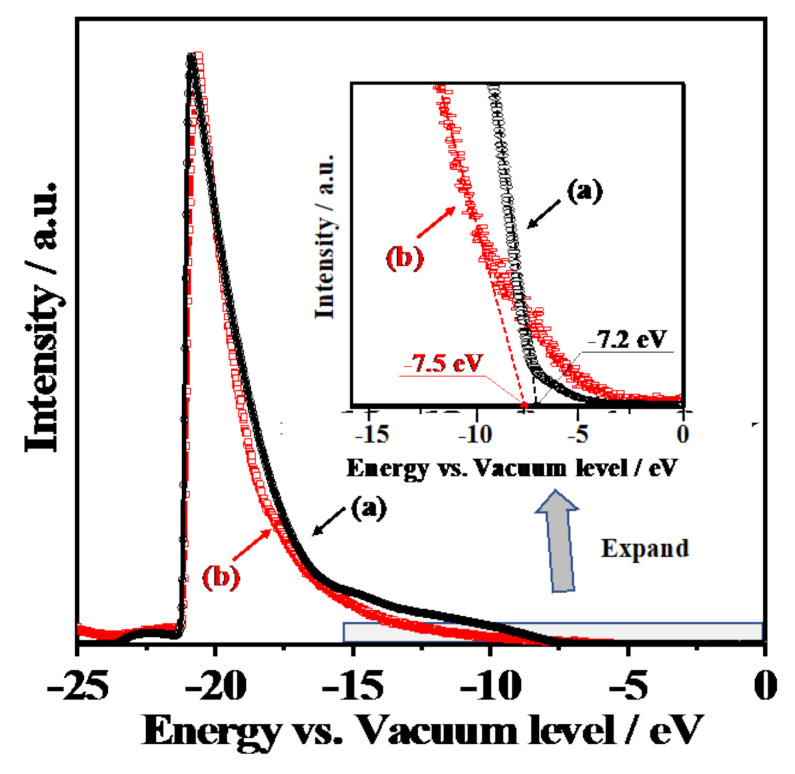
Ultra-violet photoelectron spectra (UPS) of 1573K-Pure-CeO_2_ (**a**) and 1573K-CPT-30.0YDC (**b**).

**Figure 8 materials-14-00350-f008:**
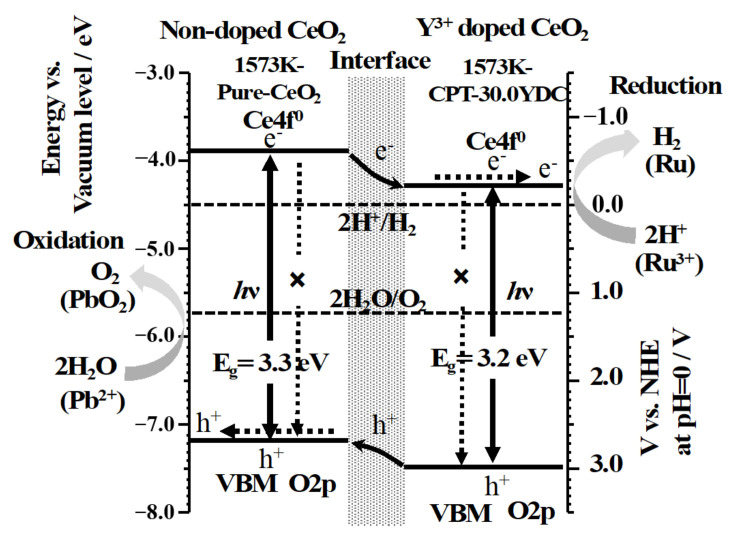
A schematic model of charge separation and efficient overall water splitting at the interface between 1573K-Pure-CeO_2_ and 1573K-CPT-30.0YDC.

**Table 1 materials-14-00350-t001:** Lattice constant of doped phase, percentage of doped Y^3+^ in CeO_2_ and percentage of doped phase.

Sample	Lattice Constant of Y^3+^-Doped Phase/Å	Percentage of Doped Y^3+^/mol %		Percentage of Doped Phase/% *^c^
		(Doped Phase) *^a^	(Average) *^b^	
1573K-Pure-CeO_2_	-	-	-	-
1573K-CPT	5.408	10	10	100
1273K-SSR	-	-	1.5	-
1373K-SSR	-	-	2.2	-
1473K-SSR	-	-	3.7	-
1573K-SSR	5.393	44	7.9	11
1673K-SSR	5.396	36	9.4	21
1773K-SSR	-	-	9.8	-

Note: *a Percentage of doped Y^3+^ (Doped phase) was calculated from their lattice constant according to references. *b Percentage of doped Y^3+^ (Average) was determined by ICP measurement. *c Percentage of doped phase was obtained by ratio of peak area for pure CeO_2_(311) and Y^3+^-doped CeO_2_(311) on the assumption that the crystallinity of both phases was the same.

**Table 2 materials-14-00350-t002:** Y^3+^ doping concentration on the surface and specific surface area.

Sample	Percentage of Doped Y^3+^ on Surface (XPS)/mol% *^a^	BET Specific Surface Area/m^2^g^−1^
**1573K-Pure-CeO_2_**	-	**4.3**
1573K-CPT	24	0.8
1273K-SSR	6	3.5
1373K-SSR	30	2.4
1473K-SSR	34	2.1
1573K-SSR	42	1.2
1673K-SSR	44	0.8
1773K-SSR	22	0.7

*a Percentage of doped Y^3+^ in CeO_2_ is calculated from XPS.

**Table 3 materials-14-00350-t003:** Photocatalytic activities for overall water splitting on RuO_2_-loaded Y^3+^-doped CeO_2_ synthesized by CPT and SSR methods at different calcination temperature with various doping amounts.

Sample	Doping Structures Determined by XRD and EDS	Activities/μmol h^−1^m^−2^	
		H_2_	O_2_
**1573K-Pure-CeO_2_**	**Non-doped**	**N.D.**	**N.D.**
1273 K-CPT	Homogeneous	N.D.	N.D.
1373 K-CPT	Homogeneous	N.D.	N.D.
1473 K-CPT	Homogeneous	N.D.	N.D.
1573 K-CPT	Homogeneous	N.D.	N.D.
1673 K-CPT	Homogeneous	N.D.	N.D.
1773 K-CPT	Homogeneous	N.D.	N.D.
1573 K-CPT-0.1YDC	Homogeneous	N.D.	N.D.
1573 K-CPT-0.5YDC	Homogeneous	N.D.	N.D.
1573 K-CPT-1.0YDC	Homogeneous	N.D.	N.D.
1573 K-CPT-5.0YDC	Homogeneous	N.D.	N.D.
1573 K-CPT-20.0YDC	Homogeneous	N.D.	N.D.
1573 K-CPT-30.0YDC	Homogeneous	N.D.	N.D.
1273 K-SSR	Heterogeneous	1.5	0.7
1373 K-SSR	Heterogeneous	5.3	2.6
1473 K-SSR	Heterogeneous	8.5	4
1573 K-SSR	Heterogeneous	30	14.4
1673 K-SSR	Heterogeneous	16.4	7.7
1773 K-SSR	Heterogeneous	3.1	1.5

## Data Availability

No new data were created or analyzed in this study. Data sharing is not applicable to this article.

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
