# Peer review of "Efficient Separation of Photoexcited Charge at Interface between Pure CeO_2_ and Y^3+^-Doped CeO_2_ with Heterogonous Doping Structure for Photocatalytic Overall Water Splitting"

_materials, 2021, doi:10.3390/ma14020350_

Round 1

Reviewer 1 Report

Saito and co-workers have presented a detailed investigation of the effect of Y3+ doping of CeO2 on the photocatalytic performance of a doped/undoped heterostructure for the water splitting reactions. The authors have provided a sufficient and clear background, and well described methodologies. The results are presented in al logical sequence, with clear graphs and tables, joined by a reasonable discussion. The conclusions are well supported with the experimental data, proving a novel and interesting approach for the understanding and development and efficient photocatalyst for water splitting.

From my point of view, I found in this manuscript novelty and good quality to be accepted for publication. I based my criteria in the following aspects:

- The authors themself have been previously reported the effects for photocatalytic water splitting of doping CeO2 with Sr and Sm. However, in the present work they went in deep on the origin of the improved photocatalytic activity in the dope/undoped CeO2 interface (using Y3+ as dopant). They have provided a detailed mechanistic discussion, supported with ultra-violet photoelectron spectroscopy data.  

- In my opinion, the introduction provides a good background for the problem faced by the authors (efficiency losses for photocatalytic water splitting due to interfacial recombination), and clearly present their strategy to better understanding and overcoming of such problematic (a mechanistic investigation for active site generation and charge separation at the dope/undoped CeO2 interface). 

- The experimental section describes in details all the procedure and reagents used by the authors. The methodology used is coherent and adequate for the purpose of the study.

- The results are presented in a logical sequence, with clear graphs and tables, which are duly discussed. In my opinion, those results sufficiently support the discussed points regarding to (i) effective preparation of Y3+ doped/undoped structures, (ii) demonstration of effective charge separation at the interface, leading to enhanced photocatalytic activity, (iii) discussion about mechanistic aspects on the role of the  dope/undoped CeO2 interface in the photocatalytic activity.  As positive aspect, I will highlight the importance of the band diagram representation (supported by experimental data) to clearly understand the role of each phase on the photocatalytic performance. 

-The final conclusions are clear and consistent with the results and discussion. 

Nonetheless, I have noticed that an English spell checking may be neccesary. I will suggest the acceptance of the manuscript, but a careful review of spelling should be made along the manuscript before publication.  

Author Response

To reviewer 1

We are very grateful to the reviewer for his/her appreciation of our work.

Comment: 1. I have noticed that an English spell checking may be neccesary. I will suggest the acceptance of the manuscript, but a careful review of spelling should be made along the manuscript before publication.

Response: Based on the comment, we have carefully reviewed and checked the entire manuscript, to resolve the confusions led by poor English writing and format. We also scrutinized the whole manuscript to improve the incoherent statements and further elevate its coherence of the manuscript.

Reviewer 2 Report

Saito et al. presented a good manuscript about the preparation of Y3+-doped CeO2 by the use of different preparation methods, demonstrating the efficent photocalytic acitivity of the doped CeO2
by CPT-methods. The authors also investigate the influence of
annealing temperature.

In my opinion the article is well written, but there are some points
to adjust:

- The authors showed the XPS characterization, but no XPS spectra are
showed, in my opinion the XPS must to be reported and discussed the
relative peaks arised from Y3+, Cerium and Oxygen.

- the SEM/TEM image are too small, also, since the authors perfomed mapping of the elements, they can calculate the % composition of the
elements and compare the results with the % obtained by the other techinques reported?

I suggest these revisions before the publication.

Author Response

To reviewer 2

Comment:1. The authors showed the XPS characterization, but no XPS spectra are showed, in my opinion the XPS must to be reported and discussed the relative peaks arised from Y3+, Cerium and Oxygen.

Response: Thanks for the reviewer’s comment. We have added the Ce3d and Y3d XPS in Figure S2. Y3+ doping concentration on surface (Table 2) were calculated by Ce3d and Y3d XPS. However, according to the carbon contamination contained with C-O-C and O-C=O (Carey IV P H, Ren F, Hays D C, et al. Valence and conduction band offsets in AZO/Ga2O3 heterostructures[J]. Vacuum, 2017, 141: 103-108.), the oxygen content may be difficult to calculate accurately by XPS.

Comment:2. The SEM/TEM image are too small, also, since the authors perfomed mapping of the elements, they can calculate the % composition of the elements and compare the results with the % obtained by the other techinques reported?

Response: Thanks for the reviewer’s comment. We enlarged the STEM and SEM images in the revised manuscript. Generally, for determining dopant concentration by EDS method, the spatial resolution is micron order. However, due to the particle size of our sample is several hundred nanometers, it is difficult to calculate reasonable dopant concentration by EDS. Thus, we used ICP, XRD and XPS instead of EDS to calculate the dopant concentration.

Reviewer 3 Report

The authors report here the synthesis of homogeneous and heterogeneous hybrid CeO2/Y3+ doped CeO2 photocatalysts for overall water splitting reaction. They showed the importance of charge transfer separation at the interface between CeO2 and Y3+ doped CeO2 phase in heterogeneous structures, especially with the increase of hydrogen and oxygen production from water (e.g. compared to the homogeneous structures).

This work is very interesting and deserves to be publish in Materials. I have though two major general concerns: first, the introduction is badly written, with misunderstandings on the hydrogen field (see the Comments section). Second, I have trouble with the differences between this work, the CeO2/Sm3+ doped CeO2, and the CeO2/Sr2+ doped CeO2 photocatalysts. I feel like the authors could better explain what is the purpose of this work compared to the previously published photocatalysts, because for what I understood, the main different point is the microscopic characterization available with Y3+ doping structures (L.81-83: “[...] it was difficult to get clear elemental images at interfaces…”).

Comments.

Introduction

L36. “Hydrogen is a zero-emission sustainable fuel…”: what the authors mean by zero emission? Do they mean no CO2 emission? In this case, the term “carbon neutral fuel” is preferred; the products of hydrogen and oxygen reaction in a fuel cell are still water and electricity…

L36-38. “hydrogen as manufactured energy […] hydrogen energy”: hydrogen is not an energy; it is described as an energy carrier. This is the main advantage of the hydrogen field, i.e. the possibility to store solar/electric energy into H2 molecules. Please correct this statement.

L38-39: “[…] the photocatalysis of TiO2…”: I hope the authors means the photocatalysis of water onto TiO2 photoelectrodes…

L44. “[…] photocatalysts still suffer from relative low efficiency…”: what the authors mean by low efficiency? In term of hydrogen/oxygen production rate, amount, of the stability of the photocatalyst in time, of the robustness of the catalyst (i.e. the hydrogen/oxygen production rate in a row with light/dark sessions)? Please precise this point.

L47. “(iii) proton reduction and water oxidation on surface.”: the proton reduction appears in acidic and neutral medium, but when basic pH aqueous solution is used, water can also be reduced in H2: 2 H2O + 2 e- = H2 + 2 HO-. I suggest to the author to use “water reduction and oxidation on surface”.

L57-59. “Li et al. reported that interface between a-Ga2O3 and b-Ga2O3 plays an important role for separation of photoexcited charge…”: One may expect citations of TiO2 Anatase/Rutile works to highlight the importance of interface between crystallographic phases, much more documented than GaO2 or other semi-conductors, especially in hydrogen photocatalysis field…

L73-75: “This discovery was characterized that the charge separation at interface between two phases gave not only activity enhancement better than each single phase but also activated expression with each inert phase for photocatalytic water splitting reaction.”: I don’t understand this sentence. Do the authors mean there is a synergy between the two phases?

Materials and Methods

L101. “under atmosphere”: Do the authors mean under air at atmospheric pressure? Under inert gas? Please precise this point.

L138. “photocatalysts were impregnated up to incipient wetness”: The authors have already used RuO2 in previous works as a support for heterogeneous photocatalysis, but do they include the charge transfer between CeO2 (doped or not) and the support as an option for increased hydrogen and oxygen production?

Results and discussion

General comment here, the use of xK-SSR-yYDC (or xK-CPT-yYDC) is very hard to follow, especially when the 10.0% of Y3+ catalyst is used in most of the publication (the molar amount of Y3+ changed only in the photocatalysis part 3.3). I suggest to the author to use a simpler name like xK-SSR (or xK-CPT) for the 10.0% Y3+ catalyst, and xK-SSR-yYDC only in the section 3.3 for photocatalysis hydrogen and oxygen production.

Syntax/spelling/conjugation

I strongly suggest to the authors to ask a native English person to read their paper, in order to avoid small mistakes and errors like the non-exhaustive list following:

  • L43. “almost of”: most of
  • L45. “was separated”: is separated
  • L50. “One way to increase the photoenergy conversion is to prevent recombination of photogenerated electron and hole. For suppressing the recombination…”: To support this purpose…
  • L51. “photocatalyst is promising approach because it acts as recombination center”: is a promising approach […] acts as a recombination…
  • L56. “is effective method”: is an effective method
  • L62. “act as driving force”: act as a driving force
  • L88. “and calcined at different calcination temperatures”: and heated at different calcination temperatures, or and calcinated at different temperatures

Author Response

To reviewer 3

Comment 1: L36. “Hydrogen is a zero-emission sustainable fuel…”: what the authors mean by zero emission? Do they mean no CO2 emission? In this case, the term “carbon neutral fuel” is preferred; the products of hydrogen and oxygen reaction in a fuel cell are still water and electricity…

Response: Thanks for the reviewer’s comment. We have rewritten this sentence as follows.

“Hydrogen is a clean and sustainable carbon neutral fuel…”

Comment 2: L36-38. “hydrogen as manufactured energy […] hydrogen energy”: hydrogen is not an energy; it is described as an energy carrier. This is the main advantage of the hydrogen field, i.e. the possibility to store solar/electric energy into H2molecules. Please correct this statement.

Response: Thanks for the reviewer’s comment. We have rewritten this sentence as follows.

“H2 and O2 is considered to be one of the most promising approaches for converting solar energy into hydrogen molecules as storage fuel.”

Comment 3: L38-39: “[…] the photocatalysis of TiO2…”: I hope the authors means the photocatalysis of water onto TiO2photoelectrodes…

Response: Thanks for the reviewer’s comment. We have rewritten this sentence as follows.

“since the photocatalytic water splitting on TiO2 photoelectrodes was discovered in 1972.”

Comment 4: L44. “[…] photocatalysts still suffer from relative low efficiency…”: what the authors mean by low efficiency? In term of hydrogen/oxygen production rate, amount, of the stability of the photocatalyst in time, of the robustness of the catalyst (i.e.the hydrogen/oxygen production rate in a row with light/dark sessions)? Please precise this point.

Response: Thanks for the reviewer’s comment. We have rewritten this sentence as follows.

“However, most of discovered photocatalysts still suffer from relative low efficiency with low hydrogen/oxygen production rate, which is not enough for practical application.”

Comment 5: L47. “(iii) proton reduction and water oxidation on surface.”: the proton reduction appears in acidic and neutral medium, but when basic pH aqueous solution is used, water can also be reduced in H2: 2 H2O + 2 e- = H2 + 2 HO-. I suggest to the author to use “water reduction and oxidation on surface”.

Response: Thanks for the reviewer’s comment. We have rewritten this sentence as follows.

“…water reduction and oxidation on surface.”

Comment 6: L57-59. “Li et al. reported that interface between a-Ga2O3 and b-Ga2O3 plays an important role for separation of photoexcited charge…”: One may expect citations of TiO2Anatase/Rutile works to highlight the importance of interface between crystallographic phases, much more documented than GaO2 or other semi-conductors, especially in hydrogen photocatalysis field…

Response: Thanks for the reviewer’s comment. We have added the reference about Anatase/Rutile- TiO2

Comment 7: L73-75: “This discovery was characterized that the charge separation at interface between two phases gave not only activity enhancement better than each single phase but also activated expression with each inert phase for photocatalytic water splitting reaction.”: I don’t understand this sentence. Do the authors mean there is a synergy between the two phases?

Response: Thanks for the reviewer’s comment. We have rewritten this sentence as follows.

“This discovery was characterized that the charge separation at interface between two phases gave activity enhancement better than each single phase, but more importantly, activated expression with each inert phase for photocatalytic water splitting reaction.”

Comment 8: L101. “under atmosphere”: Do the authors mean under air at atmospheric pressure? Under inert gas? Please precise this point.

Response: Thanks for the reviewer’s comment. We have changed “under atmosphere” to “under air at atmospheric pressure”

Comment 9: L138. “photocatalysts were impregnated up to incipient wetness”: The authors have already used RuO2 in previous works as a support for heterogeneous photocatalysis, but do they include the charge transfer between CeO2 (doped or not) and the support as an option for increased hydrogen and oxygen production?

Response: Thanks for the reviewer’s comment. RuO2 loaded undoped CeO2 and Y doped CeO2 with homogenous doping structure showed little activity for water splitting. On the other hand, RuO2 loaded Y doped CeO2 with heterogeneous doping structure exhibited high activity. The interface between doped CeO2 and undoped CeO2 facilitate the charge separation, but for water splitting, active site is necessary. Thus, we selected RuO2 to work as active site for water splitting.  

Comment 10: General comment here, the use of xK-SSR-yYDC (or xK-CPT-yYDC) is very hard to follow, especially when the 10.0% of Y3+ catalyst is used in most of the publication (the molar amount of Y3+ changed only in the photocatalysis part 3.3). I suggest to the author to use a simpler name like xK-SSR (or xK-CPT) for the 10.0% Y3+ catalyst, and xK-SSR-yYDC only in the section 3.3 for photocatalysis hydrogen and oxygen production.

Response: Thanks for the reviewer’s comment. We have changed the abbreviation as you suggested.

Comment 11: Syntax/spelling/conjugation

I strongly suggest to the authors to ask a native English person to read their paper, in order to avoid small mistakes and errors like the non-exhaustive list following:

  • L43. “almost of”: most of
  • L45. “was separated”: is separated
  • L50. “One way to increase the photoenergy conversion is to prevent recombination of photogenerated electron and hole. For suppressing the recombination…”: To support this purpose…
  • L51. “photocatalyst is promising approach because it acts as recombination center”: is a promising approach […] acts as a recombination…
  • L56. “is effective method”: is an effective method
  • L62. “act as driving force”: act as a driving force
  • L88. “and calcined at different calcination temperatures”: and heated at different calcination temperatures, or and calcinated at different temperatures

Response: Thanks for the reviewer’s comment. We have corrected the mistakes and errors you pointed out. Based on the comment, we have carefully reviewed and checked the entire manuscript, to resolve the confusions led by poor English writing and format. We also scrutinized the whole manuscript to improve the incoherent statements and further elevate its coherence of the manuscript.

Reviewer 4 Report

This work is devoted to important topic of water splitting by photocatalysts. However the novelty of work is need to be clarified. There are known examples of CeO2 dopped with Y nanoparticles and their applications (doi.org/10.1021/jp507694d, http://dx.doi.org/10.1021/jp112112u
10.1016/0167-2738(86)90111-6).

Some design changes are required:

misprints:

1) line 46, "(i)light" should be changed to "(i) light", "(ii)transfer" to "(ii) transfer"

2) line 49-50, reference is required

3) line 66 "radii" to "radius"

4) line 68-81 to big part of text without any reference

5) In general, introducation content less information about CeO2. Need to be rewritten in focus on other works devoted to CeO2 dopping.

6) fig. 2 and 3 are repetetive. It's better to leave one of them

7) need comparison of photocatalytic activity of nanoparticles obtained in this work and others

Author Response

To reviewer 4

Comment 1: There are known examples of CeO2 dopped with Y nanoparticles and their applications (doi.org/10.1021/jp507694d, http://dx.doi.org/10.1021/jp112112u
10.1016/0167-2738(86)90111-6).

Response: Thanks for the reviewer’s comment. These references were cited in our study.

Comment 2: Some design changes are required:

misprints:

1) line 46, "(i)light" should be changed to "(i) light", "(ii)transfer" to "(ii) transfer"

2) line 49-50, reference is required

3) line 66 "radii" to "radius"

4) line 68-81 to big part of text without any reference

Response: Thanks for the reviewer’s comment. The misprints 1) - 3) were corrected as you commented. For 4), a reference was added.

Comment 3: In general, introducation content less information about CeO2. Need to be rewritten in focus on other works devoted to CeO2 dopping.

Response: Thanks for the reviewer’s comment. We have added the information about CeO2 martials in introduction content. In addition, others work devoted to CeO2 doping were cited.

Comment 4: fig. 2 and 3 are repetetive. It's better to leave one of them

Response: Thanks for the reviewer’s comment. We have moved the Fig.2 (STEM/EDS mapping) to supporting information, leaving Fig.3 (FE-SEM/EDS).

Comment 5: need comparison of photocatalytic activity of nanoparticles obtained in this work and others.

Response: Thanks for the reviewer’s comment. We have compared photocatalytic activity of nanoparticles obtained in this work with others as follows:

Arakawa et al. (Bamwenda,G. R.; Sayama, K; Arakawa, H. The Photoproduction of O2 from a Suspension Containing CeO2 and Ce4+ Cations as an Electron Acceptor. Chem. Lett. 1999, 28,1047-1048.) have reported that CeO2 has an ability of O2 evolution from a suspension contained Ce4+ as an electron acceptor. On the other hand, it is reported that CeO2 suspension can produce H2using S2-/SO32- as hole acceptor (Lu, X.H.; Zhai, T.; Cui, H.; Shi, J.Y.; Xie, S.L.; Huang, Y.Y.; Liang, C.L.; Tong, Y.X. Redox cycles promoting photocatalytic hydrogen evolution of CeO2 nanorods. J. Mater. Chem. 2011, 21, 5569-5572.). The reagent like Ce4+and S2-/SO32- are called sacrificial agents. Therefore, it is desirable that photocatalytic water splitting proceed in pure water. However, no photocatalytic ability of CeO2 based materials in pure water splitting was reported except for our group, to our best knowledge.

Round 2

Reviewer 2 Report

In my opinion the XPS part are not clearly explained, no peaks are explained, no deconvolution on the spectra are present, XPS needs a large discussion, not only four lines of text.

The XPS showed as it is needs a complete revision.

In this condition, I suggest to not consider the publication of the manuscript.

Author Response

Response: Thanks for the reviewer’s comment. We have corrected the mistakes and errors pointed out from another reviewer as follows:

  • L44. “almost of”: most of
  • L46. “was separated”: is separated
  • L50. “One way to increase the photoenergy conversion is to prevent recombination of photogenerated electron and hole. For suppressing the recombination…”: To support this purpose…
  • L51. “photocatalyst is promising approach because it acts as recombination center”: is a promising approach […] acts as a recombination…
  • L56. “is effective method”: is an effective method
  • L62. “act as driving force”: act as a driving force
  • L88. “and calcined at different calcination temperatures”: heated at different calcination temperatures

Based on the comment, we have carefully reviewed and checked the entire manuscript, to resolve the confusions led by poor English writing and format. We also scrutinized the whole manuscript to improve the incoherent statements and further elevate its coherence of the manuscript.

Response: Thanks for the reviewer’s comment.

Regarding XP spectra, we have added the peak attributions and explanations about Figure S1.

For the comment of “deconvolution on the spectra”, due to Ce3d have complex satellite peaks [1-3], it is difficult to conduct “peak fitting”. Furthermore, in this research, we used XPS measurement to calculate the dopant concentration on the outmost surface. In this calculation, basing on the peak area of Ce 3d5/2, Y 3d5/2 and the corresponding Relative Sensitivity Factor in X-ray photoelectron spectroscopy (XPS; JPS-9010TR, JEOL), the percentage of doped Y3+ on surface was determined. For the determination of peak area of Y 3d5/2, because theoretically the peak area ratio of Y 3d5/2 to Y 3d3/2 is a fixed value(3d5/2 : Y 3d3/2 =3:2), the area can also be obtained without peak fitting. We revised our manuscript as follows: 

Line 199-205 : As shown in Figure S1, two peaks attributed to Y3d5/2 and Y 3d3/2 were detected on 1273K-SSR (b) and 1373K-SSR (c) in which little diffraction peaks were observed in XRD measurement. XPS measurement clearly indicates that prepared all Y3+-doped CeO2 have Y3+-doped phase on the topmost surface. Besides, Ce 3d5/2, Ce 3d3/2 and their satellite peaks were also detected in all samples. Table 2 shows Y3+ doping concentration calculated from the peak area of Ce 3d5/2, Y 3d5/2and the corresponding Relative Sensitivity Factor in X-ray photoelectron spectroscopy (XPS; JPS-9010TR, JEOL) and BET specific surface area.

Reference:

[1]. RĂDUŢOIU N, Teodorescu C M. SATELLITES IN Ce 3d X-RAY PHOTOELECTRON SPECTROSCOPY OF CERIA[J]. Digest Journal of Nanomaterials & Biostructures (DJNB), 2013, 8(4).

[2]. Mullins D R, Overbury S H, Huntley D R. Electron spectroscopy of single crystal and polycrystalline cerium oxide surfaces[J]. Surface Science, 1998, 409(2): 307-319.

[3]. Xiao W, Guo Q, Wang E G. Transformation of CeO2 (1 1 1) to Ce2O3 (0 0 0 1) films[J]. Chemical Physics Letters, 2003, 368(5-6): 527-531.

Reviewer 4 Report

Artice could be accepted as it is

Author Response

Response: Thanks for the reviewer’s comment. We have corrected the mistakes and errors pointed out from another reviewer as follows:

  • L44. “almost of”: most of
  • L46. “was separated”: is separated
  • L50. “One way to increase the photoenergy conversion is to prevent recombination of photogenerated electron and hole. For suppressing the recombination…”: To support this purpose…
  • L51. “photocatalyst is promising approach because it acts as recombination center”: is a promising approach […] acts as a recombination…
  • L56. “is effective method”: is an effective method
  • L62. “act as driving force”: act as a driving force
  • L88. “and calcined at different calcination temperatures”: heated at different calcination temperatures

Based on the comment, we have carefully reviewed and checked the entire manuscript, to resolve the confusions led by poor English writing and format. We also scrutinized the whole manuscript to improve the incoherent statements and further elevate its coherence of the manuscript.